# Divergent Driving Mechanisms Shape the Temporal Dynamics of Benthic Prokaryotic and Eukaryotic Microbial Communities in Coastal Subtidal Zones

**DOI:** 10.3390/microorganisms13051050

**Published:** 2025-04-30

**Authors:** Daode Ji, Jianfeng Zhang, Fan Li, Wensheng Li, Luping Bi, Wenlu Li, Yingjun Fu, Yunfeng Wang

**Affiliations:** 1School of Ocean, Yantai University, Yantai 264005, China; daodeji@126.com (D.J.); zjf1073355344@163.com (J.Z.); 2Shandong Provincial Key Laboratory of Restoration for Marine Ecology, Shandong Marine Resource and Environment Research Institute, Observation and Research Station of Laizhou Bay Marine Ecosystem, MNR, Yantai 264006, China; fishery.lifan@hotmail.com; 3Laizhou Mingbo Aquatic Co., Ltd., Sanshan Island Street, Yantai 261418, China; 18753565597@139.com; 4Fujian Province Key Laboratory for Coastal Ecology and Environmental Studies, College of the Environment and Ecology, Xiamen University, Xiamen 361102, China; biluping0408@163.com; 5Key Laboratory of the Ministry of Education for Coastal and Wetland Ecosystem, College of the Environment and Ecology, Xiamen University, Xiamen 361102, China; 6State Key Laboratory of Marine Environmental Science, College of Ocean and Earth Sciences, Institute of Marine Microbes and Ecospheres, Xiamen University, Xiamen 361102, China; lwl5493658@163.com (W.L.); yingjun06_26@163.com (Y.F.); 7Fujian Key Laboratory of Marine Carbon Sequestration, Xiamen University, Xiamen 361102, China

**Keywords:** coastal subtidal zones, benthic microbial communities, prokaryotes, eukaryotes, community dynamics and driving mechanisms

## Abstract

Benthic microbial communities are a vital component of coastal subtidal zones, playing an essential role in nutrient cycling and energy flow, and are fundamental to maintaining the stability and functioning of marine ecosystems. However, the response of benthic prokaryotic and eukaryotic microbial communities to environmental changes remains poorly understood. Herein, we conducted a nearly semimonthly annual sampling survey to investigate the temporal patterns and underlying mechanisms of benthic prokaryotic and eukaryotic microbial communities in the subtidal sediments of Sanshan Island, situated in the eastern Laizhou Bay of the Bohai Sea, China. The results showed that the temporal variations in benthic microbial communities followed a distinct seasonal pattern, with turnover playing a more dominant role in community succession. Nonetheless, contrasting temporal variations were observed in the alpha diversity of benthic prokaryotic and eukaryotic microbial communities, as well as in the dominant taxa across different microbial communities. Water temperature, dissolved oxygen, electrical conductivity, salinity, total nitrogen (TN), NH_4_^+^, and PO_4_^3−^ were identified as the predominant environmental drivers. The assembly of benthic microbial communities was driven by different ecological processes, in which stochastic processes mainly shaped the benthic prokaryotic communities, while deterministic processes dominated the assembly of benthic eukaryotic microbial communities. Interactions within benthic microbial communities were primarily characterized by mutualistic or cooperative relationships, but the ability of prokaryotic and eukaryotic microbial communities to maintain stability under environmental disturbances showed notable differences. These results shed light on the temporal dynamics and potential driving mechanisms of benthic prokaryotic and eukaryotic microbial communities under environmental disturbances, highlighting the distinct roles of prokaryotic and eukaryotic communities in coastal subtidal zones and providing valuable insights for the management and conservation of coastal subtidal marine ecosystems.

## 1. Introduction

As critical transitional areas between the intertidal zones and the open sea, the coastal subtidal zones are constantly influenced by the interplay of tidal forces, sediment transport, and nutrient fluxes, playing an indispensable role in sustaining biodiversity and maintaining coastal ecosystem stability [1,2]. Subtidal sediments provide a complex and heterogeneous habitat for benthic microbial communities, encompassing both prokaryotes and eukaryotes, which actively participate in organic matter degradation and nutrient transformation, contributing to the regulation of marine biogeochemical cycles [3,4]. Benthic microbial communities are highly susceptible to external environmental interference, whereas comprehensive, high-frequency, annual studies on these communities in the coastal subtidal zones, including their temporal patterns and underlying mechanisms, remain underexplored, which impedes the comprehension of how benthic microbial communities respond to rapidly varying environmental conditions [5,6]. Consequently, investigating the diversity and dynamics of benthic microbial communities in subtidal sediments through high-frequency annual sampling is essential for comprehending ecosystem functions in the coastal subtidal zones.

The intricate hydrodynamic disturbances in coastal subtidal zones can cause rapid and dramatic fluctuations in physicochemical conditions over short temporal scales, profoundly influencing the dynamics of microbial communities in benthic sediments [7]. Previous studies based on low-frequency sampling have demonstrated that Pseudomonadota dominated benthic prokaryotic communities in coastal subtidal zones; however, temporal variations in community diversity exhibited regional heterogeneity, with water temperature, nitrate, nitrite, and chlorophyll-a identified as the most important environmental factors [8,9,10]. Hitherto, most prior studies have focused on temporal patterns of benthic prokaryotic microbial communities in the coastal subtidal zones, while giving less consideration to benthic eukaryotic microbial communities. Nevertheless, eukaryotes are also vital drivers of marine biogeochemical cycles and energy flow [11]. Therefore, it is imperative to further enhance our understanding of the dynamics of benthic eukaryotic microbial communities in the coastal subtidal zones.

The mechanisms governing microbial community assembly determine diversity, succession, and biogeographic patterns, yet unraveling these processes remains a crucial but poorly understood topic in microbial ecology [12]. Neutral theory and niche theory constitute two vital and complementary mechanisms for discerning the assembly process of microbial communities [13,14]. The neutral theory insists that microbial communities are controlled by stochastic processes, such as birth, death, immigration, speciation, and limited dispersal [15]. In contrast, the niche theory considers that deterministic processes, including abiotic factors and biotic factors, shape the microbial community structure [16]. Currently, most of the previous studies concentrated on the assembly process of microbial communities in the intertidal zones and the open sea, but the assembly process of benthic microbial communities in the coastal subtidal zones is far less appreciated [17,18]. Furthermore, intraspecific and interspecific interactions within microbial communities also provide valuable insights into microbial community assembly and are important for sustaining community stability and ecosystem function [19]. Previous studies have shown that hydrodynamic changes reduce the complexity of microbial co-occurrence networks in intertidal sediments, while the co-occurrence patterns of benthic microbial communities in coastal subtidal zones remain largely unexplored [20].

In order to bridge the aforementioned research gaps, we conducted an almost semimonthly annual sampling survey to investigate the temporal patterns and underlying driving mechanisms of benthic prokaryotic and eukaryotic microbial communities in the coastal subtidal zones of Sanshan Island, located in the eastern Laizhou Bay of the Bohai Sea, China. The Sanshan Island coast experiences frequent and complex hydrodynamic disturbances, combined with strong land–ocean interactions, making it an ideal coastal subtidal area for exploring benthic microbial communities [21,22]. Given that the roles of prokaryotes and eukaryotes in natural ecosystems are well recognized due to their distinct ecological functions and adaptability to diverse environments, it is reasonable to expect that the temporal dynamics of benthic prokaryotic and eukaryotic microbial communities in the coastal subtidal zone may vary across different periods [23,24]. Utilizing a high-throughput sequencing approach, this study aims to (1) reveal and compare the diversity, community composition, and temporal variations of benthic prokaryotic and eukaryotic microbial communities and identify the vital environmental driving factors; (2) quantify the roles of stochastic processes and deterministic processes in the benthic prokaryotic and eukaryotic microbial community assembly; and (3) explore the co-occurrence patterns of benthic prokaryotic and eukaryotic microbial communities in the coastal subtidal zones. This study will be critical to understanding the response of benthic prokaryotic and eukaryotic microbial communities to environmental variations, providing a scientific basis for the conservation and sustainable management of coastal subtidal marine ecosystems, and contributing to a more comprehensive view of benthic ecosystem functioning and resilience.

## 2. Materials and Methods

### 2.1. Study Area, Sampling, and Environmental Factors

Sanshan Island is located in the eastern part of Laizhou Bay, one of the three major bays in the Bohai Sea, China. The surrounding sea area is influenced by the reciprocating tidal currents, which shape the seabed topography [25]. The sampling sites were situated in the nearshore subtidal zones of Sanshan Island, and sediment samples were collected semimonthly for ecological research on benthic prokaryotic and eukaryotic microbial communities from 11 September 2020 to 27 August 2021 (Appendix A). Three surface (0–10 cm) sediment sample replicates were collected at each site using a grab sampler, thoroughly mixed to form a composite sample, and stored in 50 mL sterile polypropylene tubes. The composite samples were kept on ice in a portable incubator and immediately transported to the laboratory. Subsequently, all sediment samples were divided into two subsamples: one was stored at 4 °C for physicochemical analysis, and the other was kept at −80 °C for DNA extraction. However, we were unable to collect some sediment samples due to irresistible circumstances, and a total of 35 surface sediment samples were obtained.

The water temperature (Temp), dissolved oxygen (DO), electrical conductivity (EC), pH, and salinity (Sal) were measured in situ using a multi-parameter water quality probe (YSI, Professional Plus, OH, USA). Total carbon (TC), total organic carbon (TOC), total inorganic carbon (TIC), and total nitrogen (TN) in the sediment interstitial water were determined by a total organic carbon analyzer. Nitrate (NO_3_^−^), nitrite (NO_2_^−^), ammonium (NH_4_^+^), phosphate (PO_4_^3−^), and metasilicate (SiO_3_^2−^) in the sediment interstitial water were measured using a continuous flow analyzer. Sulfate (SO_4_^2−^) in the sediment interstitial water was determined by ion chromatography.

### 2.2. DNA Extraction, PCR Amplification, and Sequencing

Microbial DNA was extracted from the surface sediment samples using the E.Z.N.A.^®^ Soil DNA Kit (Omega Bio-tek, Norcross, GA, USA) according to the manufacturer’s protocols. The hypervariable V4 region of the 16S and 18S rRNA gene was amplified using the primers ArBa515F (5′-GTGYCAGCMGCCGCGGTAA-3′) and 806R (5′-GGACTACHVGGGTWTCTAAT-3′) as well as Euk 528F (5′-GCGGTAATTCCAGCTCCAA-3′) and 706R (5′-AATCCRAGAATTTCACCTCT-3′), respectively [26,27]. The PCR reactions were performed in 20 μL volumes containing 4 μL of 5× FastPfu buffer, 2 μL of dNTPs (2.5 mM), 0.8 μL of each primer (5 μM), 0.4 μL of FastPfu Polymerase, and 10 ng of template DNA. The PCR reactions of the 16S amplicon included an initial denaturation at 95 °C for 5 min, followed by 28 cycles of 30 s at 95 °C, 30 s at 55 °C, 45 s at 72 °C, and a final extension of 10 min at 72 °C. The same PCR program was used for the 18S amplicon, except that 30 cycles were applied. The PCR products were visualized by 2% agarose gels and then purified using the AxyPrep DNA Gel Extraction Kit (Axygen Biosciences, Union City, CA, USA) following the manufacturer’s instructions. Purified PCR products were pooled in equimolar amounts and sequenced on the Illumina NovaSeq 6000 platform (LingEn Biotechnology Co., Ltd., Shanghai, China) using a paired-end (2 × 250 bp) approach.

### 2.3. Sequence Analysis

Raw paired-end sequencing data were demultiplexed and analyzed in QIIME2 (version 2020.11) using default parameters, where sequences were denoised, assembled with the DADA2 plugin, and clustered into amplicon sequence variants (ASVs) at a 100% identity threshold [28]. Taxonomic assignment of 16S and 18S ASVs was blasted against the Silva 16S rRNA database (release 138, http://www.arb-silva.de, accessed on 28 September 2024) and the PR^2^ 18S rRNA database [29]. ASVs that could not be taxonomically assigned, along with singletons, were discarded, and only bacterial and eukaryotic ASVs were retained for further analysis. To minimize potential biases resulting from differences in sequencing depth, all samples were rarefied to even sequencing depth (47,282 sequences/sample for bacteria and 42,955 sequences/sample for microeukaryotes) by random resampling for downstream analyses.

### 2.4. Analysis of Community Assembly Mechanisms

To unravel the potential contribution of stochastic and deterministic processes to the assembly patterns of prokaryotic and eukaryotic microbial communities, the neutral community model (NCM) was conducted to predict the relationship between the occurrence frequency of ASVs and their relative abundance across the metacommunity [30]. The parameter R^2^ in the NCM evaluates the goodness of fit to the model, whereas m represents the immigration rate. An R^2^ value close to 1 suggests that community assembly is entirely driven by stochastic processes, whereas an R^2^ value of 0 or less indicates that the community assembly does not conform to the NCM [31]. Moreover, modified stochasticity ratio (MST) analysis was also performed using the “NST” package to quantify the relative contributions of stochastic and deterministic processes to community assembly [32]. The MST values range from 0 to 1, with 1 signifying that community assembly is completely dominated by stochasticity and 0 suggesting complete dominance of determinism in the assembly process. A value of 0.5 was set as the boundary point between more stochastic (>0.5) and more deterministic (<0.5) assembly.

### 2.5. Co-Occurrence Network Analysis

To explore the co-occurrence patterns of prokaryotic and eukaryotic microbial communities, co-occurrence networks were constructed based on Spearman correlation analysis [33]. Only ASVs present in more than 25% of the samples were selected for network construction. Co-occurrence networks were constructed based on robust links (|r| > 0.7) and statistical significance (*p* < 0.05) using the “igraph” package, and network visualization was performed using Gephi v0.9.2. Network topological properties, including average degree (AD), modularity (MD), clustering coefficient (CC), average shortest path length (APL), graph density (GD), and network diameter (ND), were calculated, and higher topological characteristic values indicate a more complex network [34]. Furthermore, 1000 Erdös–Réyni random networks were generated to evaluate the randomness of the co-occurrence networks, with each random network having the identical number of nodes and edges as the actual co-occurrence network [35]. The ‘small-world’ coefficient (r) was analyzed following a previous study, where r > 1 signifies ‘small-world’ properties, reflecting high interconnectivity and efficiency [36,37].

The topological role of each node was quantified by its within-module connectivity (*Zi*) and among-module connectivity (*Pi*) [38]. Network nodes were classified into different categories: peripherals (*Zi* < 2.5 and *Pi* < 0.62), connectors (*Zi* < 2.5 and *Pi* ≥ 0.62), module hubs (*Zi* ≥ 2.5 and *Pi* < 0.62), and network hubs (*Zi* ≥ 2.5 and *Pi* ≥ 0.62) [39]. Connectors, module hubs, and network hubs are regarded as potential keystone taxa based on their vital roles in network topology.

### 2.6. Statistical Analysis

All statistical analyses were conducted using the R-Studio interface to R (version 3.6.3) and visualized by the “ggplot2” package. Alpha diversity indices, including Shannon diversity and observed ASVs, were calculated based on ASVs using the “vegan” package, and shared and unique ASVs among four seasons were showed by Venn diagrams. Non-metric multidimensional scaling (NMDS) analysis based on Bray–Curtis dissimilarity distance matrix was implemented to illustrate differences in microbial community structure between different groups. Subsequently, significant differences in microbial community structure were evaluated by the permutational multivariate analysis of variance (Adonis test) based on 999 permutations. Beta diversity partitioning of microbial communities was analyzed to distinguish nestedness and turnover components using the “betapart” package [40,41]. Canonical correspondence analysis (CCA), based on the Hellinger-transformed relative abundance matrix and environmental factors, was carried out to identify the key environmental driving factors, and the significance of each environmental factor was determined by the envfit test based on 999 permutations. Pearson correlations between environmental factors and the top 15 phyla, as well as between environmental factors and alpha diversity indices, were calculated using the “psych” package. Average variation degree (AVD) was used to evaluate the stability of microbial communities across different seasons, where a lower AVD value indicates higher community stability [42]. The significance of differences in environmental factors, the relative abundance of dominant phyla, and AVD values across different seasons were tested using the Kruskal–Wallis test.

## 3. Results

### 3.1. Temporal Dynamics of Environmental Factors

During the sampling period, most measured environmental factors showed significant seasonal variations (*p* < 0.05), except for pH, TC, TOC, NO_2_^−^, and SiO_3_^2−^ (Appendix A). The highest water temperature (23.95 ± 1.91 °C) was recorded in summer, while the lowest water temperature (4.10 ± 1.69 °C) was observed in winter. A similar seasonal pattern was also detected in the NH_4_^+^ concentration of the sediment interstitial water, with the highest concentration (11.36 ± 5.44 mg/L) in summer and the lowest (0.95 ± 0.27 mg/L) in winter. However, the concentration of DO showed an opposite trend, with the highest concentration occurring in winter (11.42 ± 0.96 mg/L) and the lowest concentration in summer (5.45 ± 0.84 mg/L). Several nearly identical temporal variations were observed among different environmental factors. The highest values of EC, Sal, TC, TIC, and SiO_3_^2−^ appeared in summer, while the lowest values were recorded in autumn. In contrast, the concentrations of TOC, PO_4_^3−^, and SO_4_^2−^ exhibited the opposite pattern, being highest in autumn and lowest in summer. Moreover, pH, NO_3_^−^, and NO_2_^−^ exhibited nearly synchronous fluctuations, peaking in spring and reaching their lowest values in autumn. On the contrary, the highest concentration of TN appeared in autumn, while the lowest occurred in spring.

### 3.2. Alpha and Beta Diversity of Benthic Microbial Communities

A total of 1,654,870 and 1,503,425 high-quality 16S and 18S rRNA gene sequences from 35 samples were classified into 10,071 and 8619 ASVs, respectively. The Shannon diversity and the number of observed ASVs in benthic prokaryotic microbial communities were 5.32 ± 0.56 and 1061 ± 218, respectively. By contrast, the Shannon diversity and the number of observed ASVs in benthic eukaryotic microbial communities were 4.11 ± 0.74 and 633 ± 134, respectively. The alpha diversity indices of benthic prokaryotic and eukaryotic microbial communities exhibited significant seasonal variations (*p* < 0.05), yet their annual variation patterns differed considerably (Figure 1A–D). The alpha diversity of prokaryotic microbial communities was at its lowest in spring, gradually increased through summer and autumn, and then declined in winter (Figure 1A,C). However, an entirely opposite trend was observed in eukaryotic microbial communities, where alpha diversity decreased from spring to autumn but significantly increased in winter (Figure 1B,D). Seasonal differences became more apparent when considering the number of shared and unique ASVs, with 12.28% to 22.45% of ASVs being unique to each season in benthic prokaryotic microbial communities and 17.26% to 18.89% in benthic eukaryotic microbial communities (Figure 1E,F).

NMDS analysis detected a clear seasonal succession in benthic prokaryotic and eukaryotic microbial communities, characterized by distinct and significant clustering of samples corresponding to individual seasons (Figure 2A,B). The Adonis test further confirmed that seasonal variations significantly affect the microbial community structure in benthic microbial communities (*p* < 0.05, Appendix A). Compared to eukaryotic microbial communities, seasonal succession in prokaryotic microbial communities was more pronounced (R^2^ = 0.244, *p* = 0.001 vs. R^2^ = 0.277, *p* = 0.001). In addition, beta diversity partitioning analysis uncovered that turnover, rather than nestedness, played a more dominant role in community succession, particularly in benthic eukaryotic microbial communities (Figure 2C,D).

### 3.3. Community Composition of Benthic Microbial Communities

The relative abundance of microbial communities was investigated to further assess the temporal variations in the composition of benthic prokaryotic and eukaryotic microbial communities. At the phylum level, the benthic prokaryotic microbial communities were dominated by Pseudomonadota (42.01 ± 9.33%), followed by Bacteroidota (27.57 ± 6.81%), Thermodesulfobacteriota (4.39 ± 1.75%), Actinomycetota (4.33 ± 1.86%), Myxococcota (4.09 ± 1.19%), Acidobacteriota (3.52 ± 1.38%), and Planctomycetota (3.22 ± 1.34%) (Figure 3A). In the benthic eukaryotic microbial communities, Alveolata (47.01 ± 17.11%), Opisthokonta (31.29 ± 19.23%), Stramenopiles (9.41 ± 5.04%), Rhizaria (6.73 ± 3.78%), and Chlorophyta (4.50 ± 3.28%) were the five predominant phyla, collectively accounting for nearly 99% of the total eukaryotic microbial sequences (Figure 3B). The composition of benthic microbial communities varied greatly at the phylum level, with most phyla exhibiting statistically significant differences in their relative abundance across different seasons (*p* < 0.05, Figure 3C,D). Intriguingly, the seasonal variations in the relative abundance of the two most abundant phyla in both prokaryotic and eukaryotic microbial communities showed a significantly opposite pattern (*p* < 0.05). Moreover, temporal changes in microbial community composition were more pronounced at the genus level than at the phylum level (Appendix A).

### 3.4. Influence of Environmental Factors on Benthic Microbial Communities

To determine the main environmental driving factors affecting benthic microbial communities, CCA and Pearson correlation analyses were applied to assess the relative importance of environmental factors in shaping microbial community structures. CCA analysis at the ASV level demonstrated significant seasonal variations in benthic microbial community structure, with the first two axes explaining 21.56% and 20.76% of the total variance in prokaryotic and eukaryotic microbial communities, respectively (Figure 4). The seasonal variations of the benthic prokaryotic microbial communities were significantly associated with nine environmental factors, with water temperature, DO, and TN being the most important environmental drivers (Figure 4A). For benthic eukaryotic microbial communities, thirteen environmental factors significantly influenced community changes, where water temperature, DO, EC, Sal, TN, NH_4_^+^, and PO_4_^3−^ were the predominant environmental drivers (Figure 4B). Pearson correlation analysis was conducted to further reveal the impact of environmental factors on alpha diversity. Similar to the results of the CCA analysis, the alpha diversity of benthic eukaryotic microbial communities was significantly affected by more environmental factors (Appendix A), implying the potential role of deterministic processes in the assembly of benthic eukaryotic microbial communities. Pearson correlation analysis was also implemented to explore the response of benthic microbial communities to environmental variations. Environmental factors were primarily significantly negatively correlated with benthic prokaryotic microbial communities, while they were mainly significantly positively associated with benthic eukaryotic microbial communities (Appendix A). Furthermore, the effects of environmental factors on the two most abundant phyla in benthic microbial communities were almost entirely opposite. For example, EC, TIC, and NO_3_^−^ were significantly positively correlated with Alveolata and significantly negatively correlated with Opisthokonta, whereas TN, PO_4_^3−^, and SO_4_^2−^ were significantly negatively associated with Alveolate and significantly positively associated with Opisthokonta.

### 3.5. Assembly Process of Benthic Microbial Communities

The relationship between the occurrence frequency and relative abundance of benthic prokaryotic microbial taxa was well described by the neutral community model (R^2^ = 0.797, Figure 5A), which is consistent with the results of modified stochasticity ratio (MST) analysis, where a high MST value of 0.674 was estimated for benthic prokaryotic microbial communities (Figure 5F). These results indicated that the assembly of benthic prokaryotic microbial communities was primarily driven by stochastic processes. However, the neutral community model provided a relatively lower fit for benthic eukaryotic microbial communities (R^2^ = 0.541, Figure 5G), and the MST value was notably lower than 0.5 (0.299, Figure 5L), which suggests that deterministic processes predominantly governed the assembly of benthic eukaryotic microbial communities. The relative importance of stochastic and deterministic processes varies across different seasons, with stochastic processes generally exerting a more pronounced influence on benthic prokaryotic and eukaryotic microbial communities in autumn (Figure 5B–F,H–L). Compared to benthic eukaryotic microbial communities, the migration rate of benthic prokaryotic microbial communities was relatively higher (Figure 5A–E,J–K).

### 3.6. Co-Occurrence Networks and Community Stability of Benthic Microbial Communities

The co-occurrence networks were constructed to explore the potential interactions and co-occurrence patterns of benthic microbial communities. The topological properties of benthic microbial co-occurrence networks, such as clustering coefficient (CC), average shortest path length (APL), and network diameter (ND), were notably higher than those of the corresponding Erdös–Rényi random networks (Appendix A), which indicates that the co-occurrence networks were non-randomly structured. The modularity of co-occurrence networks was greater than 0.6, with r values exceeding 13 (Appendix A), implying a typical modular configuration and ‘small-world’ properties of the co-occurrence networks. The benthic prokaryotic microbial co-occurrence network consisted of 710 nodes and 5014 edges, with six major modules accounting for 91.97% of the network (Figure 6A). Among the six major modules, 75.65% of the nodes were assigned as Pseudomonadota, Bacteroidota, Thermodesulfobacteriota, Acidobacteriota, Actinomycetota, and Planctomycetota (Appendix A). The proportion of positive correlations between prokaryotic taxa was 87.99%, while 12.01% showed negative correlations (Appendix A). For the benthic eukaryotic microbial co-occurrence network, the network comprised 452 nodes and 202 edges, with eight major modules accounting for 83.67% of the network (Figure 6B). Among the eight major modules, 97.04% of the nodes were classified as Alveolata, Opisthokonta, Stramenopiles, Chlorophyta, and Rhizaria (Appendix A). Analogously, the proportion of positive correlations between eukaryotic taxa was more prevalent than negative correlations (94.03% vs. 5.97%, Appendix A).

The keystone taxa in the co-occurrence networks were evaluated based on their within-module connectivity (*Zi*) and among-module connectivity (*Pi*). In the benthic prokaryotic microbial co-occurrence network, one network hub, five module hubs, and thirty-nine connectors were identified, with keystone taxa primarily from Pseudomonadota (seventeen ASVs), Bacteroidota (seven ASVs), and Actinomycetota (six ASVs) (Figure 6C, Appendix A). In the benthic eukaryotic microbial co-occurrence network, two network hubs and three connectors were detected, in which two ASVs belonged to Stramenopiles and one ASV belonged to Alveolata, Opisthokonta, and Chlorophyta (Figure 6D, Appendix A).

Additionally, the stability of microbial communities across different seasons was assessed by average variation degree (AVD). For benthic prokaryotic microbial communities, the AVD value was lower in spring, significantly increased during summer and autumn, and then notably declined in winter (Figure 6E). On the contrary, for benthic eukaryotic microbial communities, the AVD value slightly increased from spring to summer, significantly decreased in autumn, and then significantly increased in winter (Figure 6F).

## 4. Discussion

The coastal subtidal zones represent dynamic and ecologically significant habitats that play a fundamental role in marine biogeochemical cycles. As key components of coastal subtidal zones, benthic microbial communities are actively involved in nutrient transformation and elemental cycling, which contributes to the overall functioning and stability of coastal environments [43,44]. Despite their ecological importance, the temporal dynamics and underlying driving mechanisms of benthic prokaryotic and eukaryotic communities in subtidal sediments remain poorly understood. In particular, the strong hydrodynamic variability in coastal regions increases environmental fluctuations, which can significantly influence microbial diversity, succession, and co-occurrence patterns, making it difficult to accurately disclose the temporal dynamics of microbial communities based on only a few discrete time-series samples [45,46]. Thus, through an almost semimonthly annual sampling survey, this study provides the first insights into the temporal patterns and assembly mechanisms of microbial communities in response to environmental changes in the subtidal sediments of Sanshan Island, located in the eastern Laizhou Bay of the Bohai Sea, China. This study is essential for revealing the ecological functions of benthic microbial communities and predicting the stability of subtidal ecosystems under complex environmental disturbances.

### 4.1. Divergent Responses of Benthic Prokaryotic and Eukaryotic Microbial Communities to Environmental Factors

The dynamic changes in diversity and species composition of microbial communities could reflect their responses to environmental disturbances, offering insight into the stability and adaptability of the ecosystem [47,48]. In this study, the diversity and composition of benthic microbial communities exhibited distinct seasonal variations in response to fluctuating environmental factors, which reflect the self-adaptation of ecosystems to the changing environmental conditions [49]. The significant seasonal dynamics observed in benthic microbial communities are supported by findings from previous research on various coastal ecosystems [50,51,52]. Water temperature, DO, EC, Sal, and nutrients (TN, NH_4_^+^, PO_4_^3−^) were found to be the most important environmental factors affecting the succession of benthic microbial communities, which may be attributed to the fact that these environmental factors could directly influence the growth and reproduction of microbial communities by regulating their substrate utilization and metabolic processes [53,54]. Compared to nestedness, turnover exerted a more significant influence on the succession of benthic microbial communities, highlighting the importance of conserving the biodiversity of benthic microbial habitats across coastal regions [55].

It is worth noting that the changes in alpha diversity of benthic prokaryotic and eukaryotic microbial communities exhibited completely opposite temporal patterns, particularly that of the eukaryotic microbial communities which significantly decreased in autumn. One possible explanation for this difference may lie in the contrasting responses and adaptive capacities of prokaryotes and eukaryotes to environmental disturbances. In recent years, extreme weather phenomena, particularly typhoons and extreme rainfall events, triggered abnormal autumn floods in Laizhou Bay, decreasing seawater salinity and intensifying coastal hydrodynamic fluctuations, which significantly impacted microbial communities [56,57]. The flexible metabolic potential of prokaryotes enables them to rapidly respond to changing environmental conditions, whereas the complex life cycles and relatively lower environmental adaptability of eukaryotes diminish their capacity to cope with sudden anomalous events, which may explain the significant decline in eukaryotic microbial alpha diversity in autumn [58,59].

Pseudomonadota and Bacteroidota were the dominant phyla in the benthic prokaryotic microbial communities, while Alveolata and Opisthokonta were predominant in the benthic eukaryotic microbial communities. This observation is consistent with previous studies conducted in various marine ecosystems, albeit with variations in their relative abundance [60,61]. The dominance of these phyla in coastal sediments may be related to their competitive advantage, stemming from their high resource utilization efficiency and adaptability to external conditions [62]. Nonetheless, the dominant taxa within benthic microbial communities exhibited entirely different seasonal variation patterns. This discrepancy might be owing to the contrasting effects of environmental factors on them, as confirmed by correlation analysis (Appendix A). Furthermore, the predation and competition relationships among microorganisms may also be a significant factor contributing to this discrepancy [63,64].

### 4.2. The Relative Importance of Deterministic and Stochastic Processes Varied Between Benthic Prokaryotic and Eukaryotic Microbial Communities

Deciphering the relative importance of ecological processes driving the microbial community assembly has become a central goal in microbial ecology [65]. In this study, the assembly process of benthic prokaryotic microbial communities was dominated by stochastic processes. Despite limited research on the assembly of prokaryotic microbial communities in coastal subtidal zones, previous studies in estuarine, intertidal, and marine regions supported the predominant role of stochastic processes in shaping prokaryotic community assembly [18,66,67]. Contrarily, deterministic processes primarily shaped the benthic eukaryotic microbial communities, as further confirmed by CCA analysis (Figure 4), implying that the assembly process of benthic eukaryotic microbial communities was significantly influenced and filtered by environmental factors, which is in accordance with previous findings in various ecosystems [68,69,70,71]. Two possible reasons may explain the contrasting roles of deterministic and stochastic processes in benthic prokaryotic and eukaryotic microbial communities. Firstly, compared to eukaryotic microbial communities, prokaryotic microbial communities demonstrate a higher migration capacity, which facilitates their random dispersal, resulting in a more pronounced influence of stochastic processes on benthic prokaryotic communities [45,68]. Secondly, compared to prokaryotes, eukaryotes possess more complex cellular structures and respond more extensively to environmental heterogeneity, which renders deterministic processes more important in shaping benthic eukaryotic microbial community assembly [54]. In addition, the relative importance of stochastic processes clearly increased in autumn, which may be ascribed to the abnormal autumn floods that intensified coastal hydrodynamic fluctuations, increased the occurrence of diffusion events, and led to a higher level of compositional stochasticity and variability, thereby amplifying the role of stochastic processes [16,34].

### 4.3. Co-Occurrence Patterns and Seasonal Variations in the Community Stability of Benthic Prokaryotic and Eukaryotic Microbial Communities

Co-occurrence network analysis provides key evidence for revealing microbial co-occurrence patterns, in which microbial interactions could impact community composition and regulate the stability and distribution patterns of microbial communities [72,73]. In this study, typical modular configurations and ‘small-world’ properties were observed in the co-occurrence networks, indicating complex and stable interactions among the benthic microbial communities [74]. The proportion of positive correlations between microbial taxa was more prevalent than negative correlations, reflecting that the benthic microbial communities were dominated by mutualistic or cooperative relationships, such as co-aggregation, cross-feeding, and co-colonization [19,75]. Compared to the benthic eukaryotic microbial co-occurrence network, the benthic prokaryotic microbial co-occurrence network showed higher average degree, clustering coefficient, and negative correlation ratio, along with more keystone taxa and a lower average shortest path length, suggesting that the benthic prokaryotic microbial communities had more complex and stable interaction patterns [76]. Keystone taxa hold a disproportionate role in maintaining the co-occurrence network structure, and their disappearance could lead to the collapse of the co-occurrence network [77,78]. The keystone taxa in the benthic microbial co-occurrence networks primarily belong to the phyla Pseudomonadota, Bacteroidota, Actinomycetota, Stramenopiles, Alveolata, Opisthokonta, and Chlorophyta, which may be primarily linked to their strong environmental adaptability and their contributions to nutrient cycling and energy metabolism [17,79,80].

The analysis of average variation degree (AVD) in this study demonstrated that the stability of benthic microbial communities varied across different seasons, which aligned with previous studies conducted in various aquatic ecosystems, where seasonal variations imposed a significant influence on the stability of microbial co-occurrence networks [75,76,81]. In our study, the annual variations in the stability of benthic prokaryotic and eukaryotic microbial communities were almost identical, with the exception of autumn. The relatively high hydrological connectivity and fluctuations induced by abnormal autumn floods, coupled with the distinct environmental responses of prokaryotes and eukaryotes, may provide a plausible explanation for this phenomenon. Hydrodynamic disturbances facilitate the dispersal of sediment-attached microorganisms, with a more pronounced impact on prokaryotes [82]. This may lead to the homogenization of microbial communities and cause microbial taxa to form highly synchronous modules that resist hydrodynamic disturbances; however, the high synchrony among microbial taxa ultimately decreased the stability of benthic prokaryotic microbial communities [83]. Compared with prokaryotes, the higher sediment-attachment ability and lower migration capacity of eukaryotes could be conducive to reducing the risk of community homogenization. Additionally, eukaryotes respond more extensively to environmental changes, which may render them more likely to establish tighter associations to enhance community stability and to withstand external environmental disturbances caused by abnormal autumn floods [75,84]. This might further contribute to the deterministic processes of benthic eukaryotic microbial community assembly in autumn by serving as a form of selection force, i.e., niche differentiation [85,86].

## 5. Conclusions

To the best of our knowledge, this study was the first comprehensive investigation exploring the temporal dynamics and potential driving mechanisms of benthic prokaryotic and eukaryotic microbial communities in coastal subtidal zones, based on an almost semimonthly sampling over a year. Our findings revealed that the temporal dynamics of benthic microbial diversity and community structure varied across different seasons, whereas the temporal variations in alpha diversity and dominant taxa of prokaryotic and eukaryotic microbial communities exhibited opposing patterns. Turnover, rather than nestedness, played a more dominant role in community succession. Water temperature, DO, EC, Sal, TN, NH_4_^+^, and PO_4_^3−^ were found to be the most important environmental factors affecting the succession of benthic microbial communities. Stochastic processes dominated the assembly of benthic prokaryotic microbial communities, while deterministic processes had a greater influence on the assembly of eukaryotic microbial communities. Mutualistic or cooperative relationships were predominant within the benthic microbial communities, but the stability of prokaryotic and eukaryotic microbial communities in response to environmental disturbances was different. Overall, this study deepens our understanding of benthic prokaryotic and eukaryotic microbial community responses to environmental changes, offering valuable insights for safeguarding the stability of coastal ecosystems.

## Figures and Tables

**Figure 1 microorganisms-13-01050-f001:**
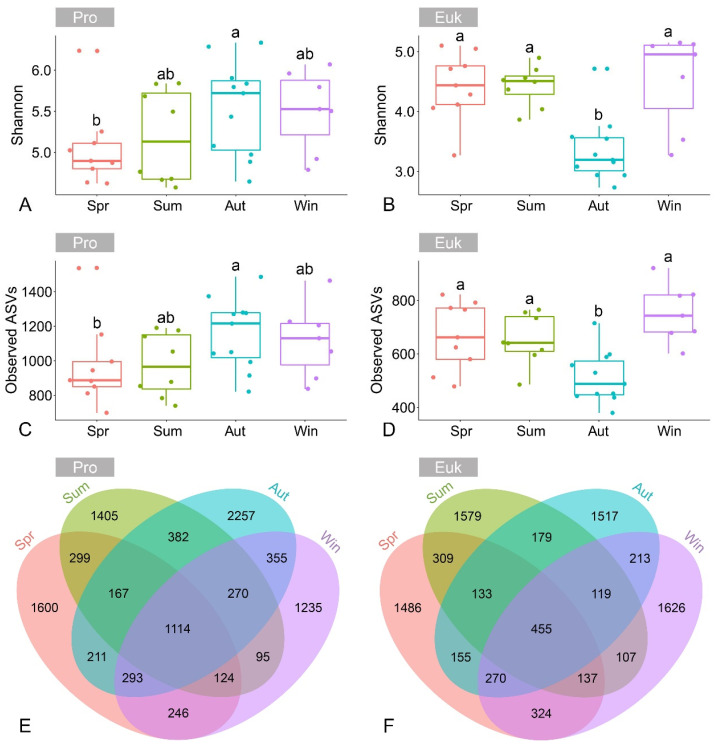
Alpha diversity indices of benthic prokaryotic (**A**,**C**) and eukaryotic (**B**,**D**) microbial communities obtained from different sampling seasons. Different letters indicate significant differences (*p* < 0.05, Kruskal–Wallis test). Venn diagrams showing the number of shared and unique ASVs among four seasons in benthic prokaryotic (**E**) and eukaryotic (**F**) microbial communities. Abbreviations: Pro, prokaryotic microbial communities; Euk, eukaryotic microbial communities; Spr, spring; Sum, summer; Aut, autumn; Win, winter.

**Figure 2 microorganisms-13-01050-f002:**
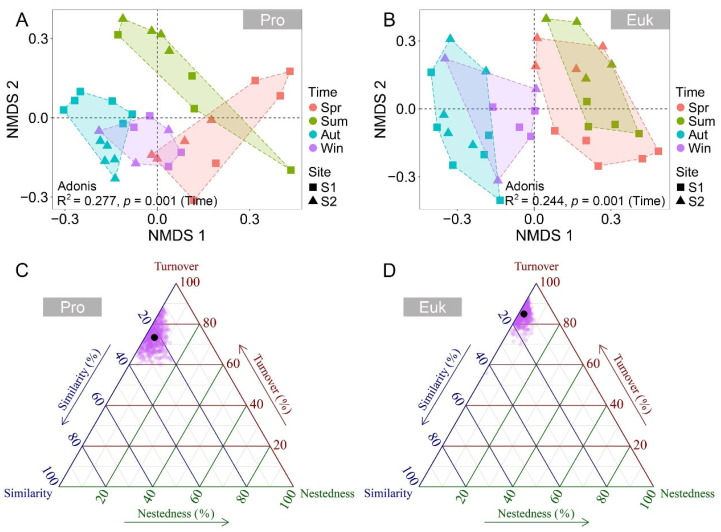
Non-metric multidimensional scaling (NMDS) ordination showing the distribution patterns of benthic prokaryotic (**A**) and eukaryotic (**B**) microbial communities among four seasons based on Bray–Curtis dissimilarities. Significance is assessed using Adonis test with 999 permutations. Triangular diagrams of beta diversity comparisons (using Jaccard dissimilarity index) for benthic prokaryotic (**C**) and eukaryotic (**D**) microbial communities. Each purple point represents a pair of samples, and its position is determined by a triplet of values from the similarity, turnover, and nestedness matrices. The mean values of similarity, turnover, and nestedness are indicated by larger black points. Abbreviations: Pro, prokaryotic microbial communities; Euk, eukaryotic microbial communities; Spr, spring; Sum, summer; Aut, autumn; Win, winter.

**Figure 3 microorganisms-13-01050-f003:**
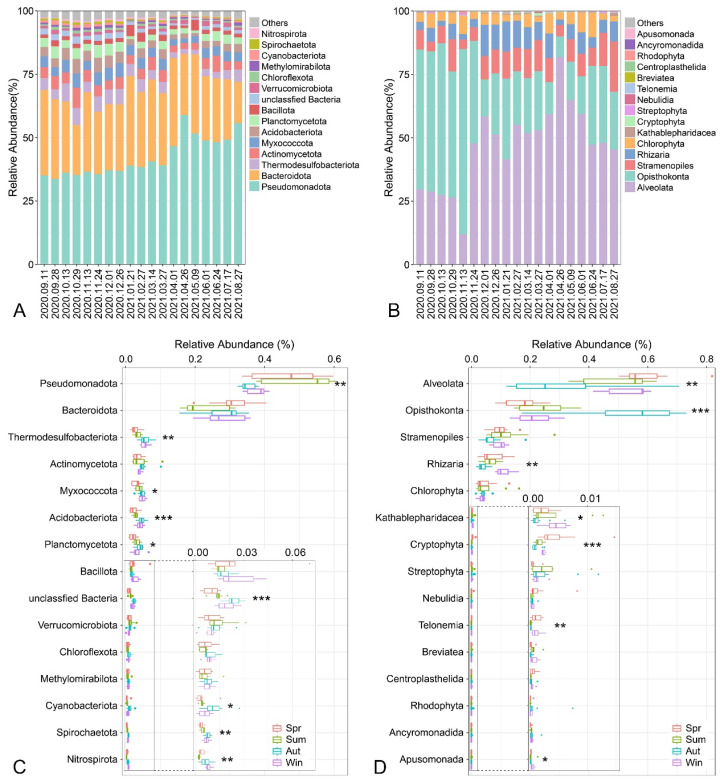
Temporal dynamics of relative abundance at the phylum level (Top 15) in benthic prokaryotic (**A**) and eukaryotic (**B**) microbial communities from 11 September 2020 to 27 August 2021. Differences in the relative abundance of dominant phyla (Top 15) in benthic prokaryotic (**C**) and eukaryotic (**D**) microbial communities across different seasons. The significance of differences in the relative abundance of dominant phyla across different seasons was tested using the Kruskal–Wallis test. Significance levels are denoted as * *p* < 0.05; ** *p* < 0.01; *** *p* < 0.001. Abbreviations: Spr, spring; Sum, summer; Aut, autumn; Win, winter.

**Figure 4 microorganisms-13-01050-f004:**
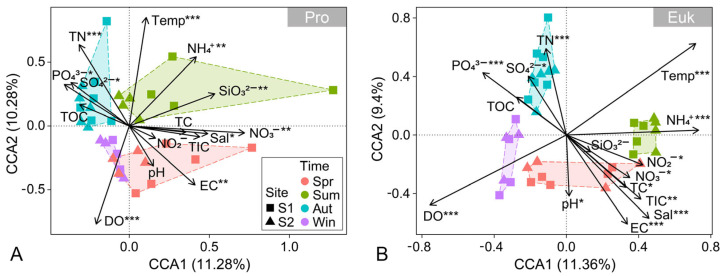
Canonical correspondence analysis (CCA) showing the benthic prokaryotic (**A**) and eukaryotic (**B**) microbial community composition in relation to environmental factors. Significance: *, *p* < 0.05; **, *p* < 0.01; ***, *p* < 0.001. Abbreviations: Pro, prokaryotic microbial communities; Euk, eukaryotic microbial communities; Spr, spring; Sum, summer; Aut, autumn; Win, winter.

**Figure 5 microorganisms-13-01050-f005:**
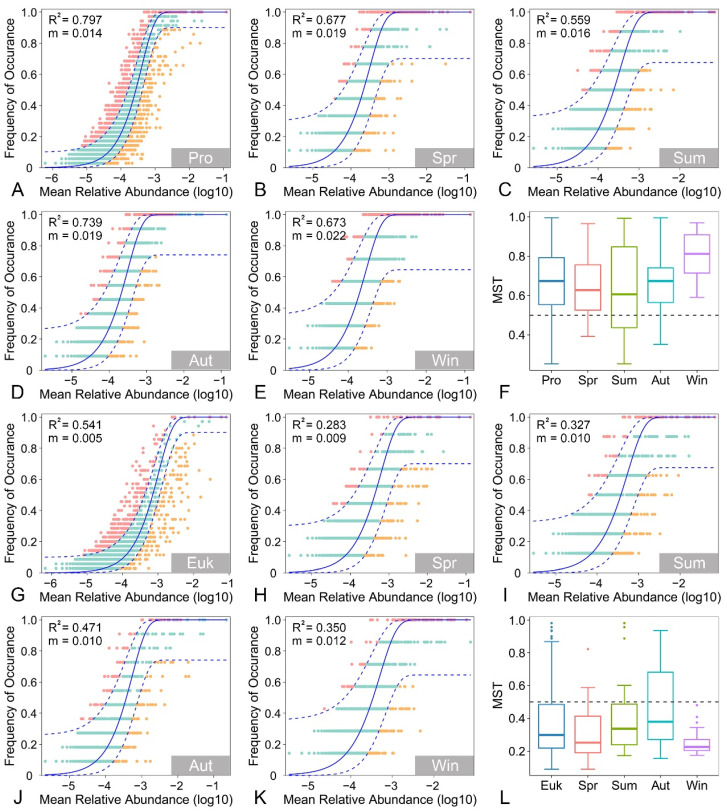
Fit of the neutral community model (NCM) for benthic prokaryotic (**A**–**E**) and eukaryotic (**G**–**K**) microbial communities. The predicted occurrence frequencies for prokaryotic and eukaryotic microbial communities across the entire year (**A**,**G**), as well as in spring (**B**,**H**), summer (**C**,**I**), autumn (**D**,**J**), and winter (**E**,**K**), respectively. The solid blue lines are the best fit to the NCM, and the dashed blue lines indicate 95% confidence intervals around the NCM prediction. ASVs that occur more or less frequently than predicted by the NCM are shown in pink and orange, respectively. R^2^ is the fit to the NCM, and m is the estimated migration rate. The modified stochasticity ratio (MST) based on Bray–Curtis method showing the null deviation of benthic prokaryotic (**F**) and eukaryotic (**L**) microbial communities across the entire year, as well as in spring, summer, autumn, and winter, respectively. Abbreviations: Pro, prokaryotic microbial communities; Euk, eukaryotic microbial communities; Spr, spring; Sum, summer; Aut, autumn; Win, winter.

**Figure 6 microorganisms-13-01050-f006:**
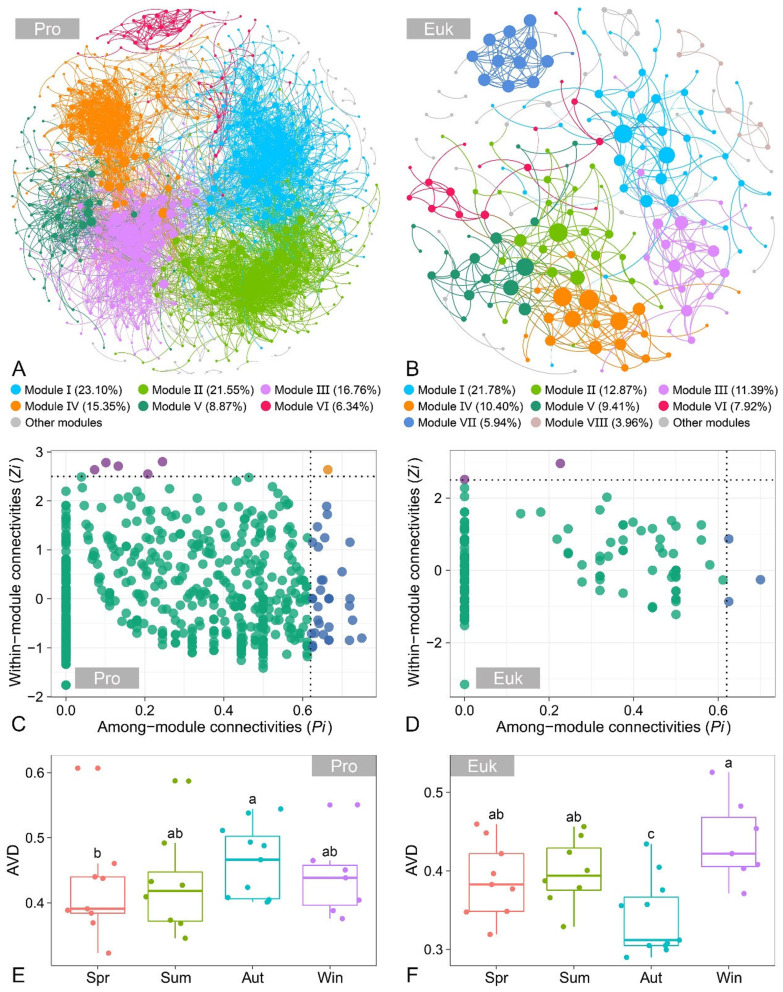
The co-occurrence networks of benthic prokaryotic (**A**) and eukaryotic (**B**) microbial communities, with nodes colored according to different modularity classes. The size of each node is proportional to its degree, and each edge represents a strong and significant correlation. The identification of keystone taxa in the benthic prokaryotic (**C**) and eukaryotic (**D**) microbial co-occurrence networks based on within-module connectivity (*Zi*) and among-module connectivity (*Pi*), with threshold values of 2.5 and 0.62. Average variation degree (AVD) of benthic prokaryotic (**E**) and eukaryotic (**F**) microbial communities across different seasons. Different letters indicate significant differences (*p* < 0.05, Kruskal–Wallis test). Abbreviations: Pro, prokaryotic microbial communities; Euk, eukaryotic microbial communities; Spr, spring; Sum, summer; Aut, autumn; Win, winter.

## Data Availability

The datasets presented in this study will be found in online repositories. The names of the repository/repositories and accession number(s) can be found from the NCBI database: PRJNA1255835.

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
