# Peer review of "Divergent Driving Mechanisms Shape the Temporal Dynamics of Benthic Prokaryotic and Eukaryotic Microbial Communities in Coastal Subtidal Zones"

_microorganisms, 2025, doi:10.3390/microorganisms13051050_

Round 1

Reviewer 1 Report

Comments and Suggestions for Authors

The authors of this substantial coastal benthic seasonal study located in the Bohai Sea, China conclude “ Through an almost semimonthly  annual sampling survey, this study provides the first insights into the temporal patterns and assembly mechanisms of microbial communities in response to environmental changes in the subtidal sediments of Sanshan Island, located in the eastern Laizhou Bay  of the Bohai Sea, China. This study is essential for revealing the ecological functions of benthic microbial communities and predicting the stability of subtidal ecosystems under  complex environmental disturbances.“

In most respects, the submitted manuscript appears to address these goals fairly substantively and comprehensively. My expertise is in physiological ecology and I am not fully expert in some of the statistical analyses used. However, the authors provide ample figural and mathematical evidence to support most of their conclusions. The sampling sites were situated in the nearshore subtidal zones of Sanshan Island, and sediment samples were collected  semimonthly for ecological research on benthic prokaryotic and eukaryotic microbial  communities from September 11, 2020 to August 27, 2021.

The study utilized  high- throughput sequencing to : (1) reveal and compare the diversity, community composition and temporal variations of benthic prokaryotic and eukaryotic microbial communities, as well as identify the vital environmental driving factors; (2) quantify the roles of stochastic processes and deterministic processes in the benthic prokaryotic and eukaryotic microbial community assembly; (3) explore the co-occurrence patterns of benthic prokaryotic and eukaryotic microbial communities in the coastal sub-tidal zones.

The comparative analysis of both prokaryotic and eukaryotic assemblages, largely using top 15 phyla each, is sufficiently representative,

I noticed only a few sentences that may be improved

 Line                Suggestion

36–37

Water temperature, dissolved oxygen, electrical conductivity, salinity, total nitrogen (TN), NH4+ and PO43- were identified as the predominant environmental drivers.   Simply spelling out the word total nitrogen  would be helpful. Most  professional readers would recognize the abbreviation TN as total nitrogen, but to be complete, the word should be spelled out first time the abbreviation is used.

492–494. Two possible reasons may explain the contrasting roles of deterministic and stochastic processes in benthic 493 prokaryotic and eukaryotic microbial communities.    The word ‘that’ is not needed.

Author Response

Comments and Suggestions from Reviewer 1

The authors of this substantial coastal benthic seasonal study located in the Bohai Sea, China conclude “Through an almost semimonthly annual sampling survey, this study provides the first insights into the temporal patterns and assembly mechanisms of microbial communities in response to environmental changes in the subtidal sediments of Sanshan Island, located in the eastern Laizhou Bay of the Bohai Sea, China. This study is essential for revealing the ecological functions of benthic microbial communities and predicting the stability of subtidal ecosystems under complex environmental disturbances.”

In most respects, the submitted manuscript appears to address these goals fairly substantively and comprehensively. My expertise is in physiological ecology and I am not fully expert in some of the statistical analyses used. However, the authors provide ample figural and mathematical evidence to support most of their conclusions. The sampling sites were situated in the nearshore subtidal zones of Sanshan Island, and sediment samples were collected semimonthly for ecological research on benthic prokaryotic and eukaryotic microbial communities from September 11, 2020 to August 27, 2021.

The study utilized  high-throughput sequencing to : (1) reveal and compare the diversity, community composition and temporal variations of benthic prokaryotic and eukaryotic microbial communities, as well as identify the vital environmental driving factors; (2) quantify the roles of stochastic processes and deterministic processes in the benthic prokaryotic and eukaryotic microbial community assembly; (3) explore the co-occurrence patterns of benthic prokaryotic and eukaryotic microbial communities in the coastal sub-tidal zones.

The comparative analysis of both prokaryotic and eukaryotic assemblages, largely using top 15 phyla each, is sufficiently representative.

R: We are very grateful to you for your recognition and high evaluation of our research.

I noticed only a few sentences that may be improved

 Line                Suggestion

36–37. Water temperature, dissolved oxygen, electrical conductivity, salinity, total nitrogen (TN), NH4+ and PO43- were identified as the predominant environmental drivers.  Simply spelling out the word total nitrogen would be helpful. Most professional readers would recognize the abbreviation TN as total nitrogen, but to be complete, the word should be spelled out first time the abbreviation is used.

R: Revised, thank you.

492–494. Two possible reasons may explain the contrasting roles of deterministic and stochastic processes in benthic 493 prokaryotic and eukaryotic microbial communities.    The word ‘that’ is not needed.

 R: Revised, thank you.

Reviewer 2 Report

Comments and Suggestions for Authors

REVISION: Divergent Driving Mechanisms Shape the Temporal Dynamics of Benthic Prokaryotic and Eukaryotic Microbial Communities in Coastal Subtidal Zones

Introduction – Conclusion Paragraph:

It is recommended to add a final paragraph to the introduction that clearly highlights the novel scientific contribution of this study, in order to emphasize the necessity and relevance of its publication.

Section 2.1 – Analytical Methods:

In Section 2.1, the measured parameters are listed (from total carbon to sulfates); however, it would be advisable to provide more detailed information on the actual analytical methods used, rather than simply mentioning the use of ion chromatography.

Statistical Analysis:

The statistical analysis represents a key aspect of this study. We suggest enhancing it by including a multiple linear regression model to explore relationships among the parameters more thoroughly.

Figure 1 – Data Visualization:

The box plots presented in Figure 1 could be replaced with violin plots, which may offer a more comprehensive visual representation of the data distribution.

Finally, we congratulate the authors on the valuable work conducted.

Author Response

Comments and Suggestions from Reviewer 2

REVISION: Divergent Driving Mechanisms Shape the Temporal Dynamics of Benthic Prokaryotic and Eukaryotic Microbial Communities in Coastal Subtidal Zones

Introduction – Conclusion Paragraph:

It is recommended to add a final paragraph to the introduction that clearly highlights the novel scientific contribution of this study, in order to emphasize the necessity and relevance of its publication.

R: Thank you for your valuable suggestion. We have supplemented the final paragraph of the introduction to clearly highlight the novel scientific contributions and significance of our study, thereby emphasizing the necessity and relevance of its publication.

Section 2.1 – Analytical Methods:

In Section 2.1, the measured parameters are listed (from total carbon to sulfates); however, it would be advisable to provide more detailed information on the actual analytical methods used, rather than simply mentioning the use of ion chromatography.

R: Thank you for your valuable suggestion. The measurements of environmental parameters in our study (e.g., total carbon, nitrate, sulfate) were performed according to the standard protocols of specific analytical instruments. These measurements primarily relied on calibration solutions of known concentrations, with environmental concentrations determined directly from the standard curves derived from these solutions. As such, the analyses did not involve complex analytical methods. Therefore, we briefly described the instruments and general analytical techniques (e.g., total organic carbon analyzer, ion chromatography) used in the manuscript, which is a commonly accepted reporting style in many molecular ecology studies.

Statistical Analysis:

The statistical analysis represents a key aspect of this study. We suggest enhancing it by including a multiple linear regression model to explore relationships among the parameters more thoroughly.

R: Thank you very much for your thoughtful suggestion. We fully agree that statistical analysis is a critical component of this study. However, we respectfully note that multiple linear regression is not universally applicable to all types of ecological or microbiome data, especially when dealing with high-dimensional, non-normally distributed, or compositional datasets, as is often the case in microbial community studies.

In this study, we employed a suite of statistical approaches—including the Adonis test, Mantel test, Kruskal-Wallis test, NMDS analysis, Pearson correlation analysis, and canonical correspondence analysis (CCA). These methods are well-established and widely applied in the fields of molecular ecology and biostatistics, and are more appropriate for capturing the complex multivariate relationships between microbial community composition and environmental factors. Therefore, we believe that the current analytical framework is appropriate and sufficient for addressing the study’s research questions.

Figure 1 – Data Visualization:

The box plots presented in Figure 1 could be replaced with violin plots, which may offer a more comprehensive visual representation of the data distribution.

R: Thank you for your valuable suggestion. The box plots used in our study also provide valuable insights into the data distribution. We have displayed numerous individual data points within the box plots, which effectively illustrate the spread of the data and serve a function similar to that of violin plots. In addition, box plots are widely used and easy to interpret, making them a more suitable choice in this context. Therefore, we have retained the box plots in the revised manuscript.

Finally, we congratulate the authors on the valuable work conducted.

R: We are very grateful to you for your recognition and high evaluation of our research.
